# Mujeres Unidas: Addressing Substance Use, Violence, and HIV Risk through Asset-Based Community Development for Women in the Sex Trade

**DOI:** 10.3390/ijerph18083884

**Published:** 2021-04-07

**Authors:** Lianne A. Urada, Andrés Gaeta-Rivera, Jessica Kim, Patricia E. Gonzalez-Zuniga, Kimberly C. Brouwer

**Affiliations:** 1College of Health and Human Services, San Diego State University School of Social Work, San Diego, CA 92182, USA; 2Department of Medicine, Division of Infectious Diseases and Global Public Health, University of California San Diego, La Jolla, CA 92093, USA; kbrouwer@ucsd.edu; 3Facultad de Medicina y Ciencias Biomédicas, Universidad Autonoma de Chihuahua, Chihuahua 31125, Mexico; andres.gaeta@gmail.com; 4Center for Justice and Reconciliation, Point Loma Nazarene d, San Diego, CA 92106, USA; jkim1@pointloma.edu; 5Casa del Centro and the Wound Clinic, Tijuana 22000, Mexico; lacasadelcentro1@gmail.com; 6Department of Family Medicine and Public Health, University of California San Diego, La Jolla, CA 92093, USA

**Keywords:** women, sex trade, substance use, HIV, violence, community mobilization, asset-based community development, empowerment, human trafficking

## Abstract

This paper examines the prevalence of and potential for community mobilization (CM) and its association with HIV/STI risk, substance use, and violence victimization among women, particularly those using substances, in the sex trade in Tijuana, Mexico. Methods: 195 women participated in Mujeres Unidas (K01DA036439 Urada) under a longitudinal survey study, “Proyecto Mapa de Salud” (R01DA028692, PI: Brouwer). Local health/social service providers (N = 16) were also interviewed. Results: 39% of women who participated in community mobilization activities used substances. In adjusted analyses (*n* = 135), participation in CM activities (*n* = 26) was more likely among women who did not report substance use (AOR: 4.36, CI: 1.11–17.16), perceived a right to a life free from violence (AOR: 9.28, CI: 2.03–59.26), talked/worked with peers in the sex trade to change a situation (AOR: 7.87, CI: 2.03–30.57), witnessed violence where they worked (AOR: 4.45, CI: 1.24–15.96), and accessed free condoms (AOR: 1.54, CI: 1.01–2.35). Forty-five of the women using substances demonstrated their potential for engaging in asset-based community development (ABCD) with service providers in Mujeres Unidas meetings. Conclusion: Women using substances, vs. those who did not, demonstrated their potential to engage in ABCD strategies. Women’s empowerment, safety, and health could be enhanced by communities engaging in ABCD strategies that build and bridge social capital for marginalized women who otherwise have few exit and recovery options.

## 1. Introduction

The 2030 Agenda for Sustainable Development [1] adopted by the United Nations (2015) aims to “achieve gender equality and to empower all women and girls globally” (Goal 5) by “eliminating all forms of violence against women and girls, including trafficking and sexual and other types of exploitation” (5.2), and “ensuring women’s full and effective participation and equal opportunities for leadership at all levels of decision-making in political, economic, and public life (5.5).”

With this in mind, our study, Mujeres Unidas, used asset-based community development (ABCD) as a sustainable development strategy to focus on community strengths and assets as a way of developing and sustaining a supportive community of marginalized women instead of focusing on deficits or needs alone [2]. Kretzmann and McKnight (1993) developed ABCD with origins in community organizing, sociology, and urban affair, and social work embraces this ABCD’s strengths-based approach to uplifting communities. Applied globally, ABCD recognizes local capacities and mobilizes citizens’ resources in lower-income communities [2]. As part of ABCD, we focused on community mobilization (CM) as a “capacity-building process through which individuals, groups, or organizations plan, carry out, and evaluate activities to improve their health and other needs” [3]. CM empowers individuals and groups to take action to affect change at policy levels (e.g., influencing government, police, health systems) via “mobilization of resources, disseminating information, generating support, and fostering cooperation across public and private sectors in the community” [4,5]. ABCD encourages a whole community approach to addressing the needs of a group by tapping into external social capital where few supportive services exist.

Related to another UN sustainable development goal to ensure healthy lives and promote the well-being of all (Goal 3) [1], we focused on examining human immunodeficiency virus and sexually transmitted infections (HIV/STIs) among women in the sex trade. This UN goal aims to “end the epidemics of AIDS and other communicable diseases (3.3), and strengthen the prevention and treatment of substance abuse, including narcotic drug abuse and harmful use of alcohol (3.5)”. Globally, women trading sex have 14 times the risk for HIV relative to same-age women in the general population [6]. However, social and structural community factors often determine HIV risk among those in the sex trade [7,8,9,10]. Women using substances in particular face greater harm and social isolation [11,12].

Women trading sex have taken collective action to reduce their risks of violence, abuse, HIV/STIs, and to increase their access to healthcare and other income [13,14,15,16,17,18,19,20,21]. For example, they have mobilized for human rights via a group, meeting, or marches, sharing concerns and helping each other. CM has reduced HIV/STI risk among women in the sex trade in India, the Caribbean, Brazil, and Africa [13,14,15,16,17,18,19,20,21]. Its dissemination in India has resulted in significant reductions in HIV prevalence at regional levels [22]. Women have built consciousness and solidarity among their peers through rights-based framing while diffusing HIV prevention messages to the community [15,19]. In the context of HIV, persons trading sex have mobilized for their occupational health and safety, e.g., through HIV prevention organizations [23,24,25,26,27]. Community mobilization has also been successful in reducing HIV among persons using substances in India and the U.S., using a harm reduction approach [28,29].

However, some of the most marginalized women, those using substances in the sex trade, are excluded from mobilization efforts, e.g., from other groups in Mexico [30]. Yet many could benefit from an ABCD approach to escape substance use, violence victimization, and harms to their health. HIV prevalence is higher among women in the sex trade who use substances (12% among those injecting drugs vs. 6% among those who did not) [31]. In a study of 924 women trading sex in Tijuana, one in five ever injected drugs and 16% injected in the past month [32,33].

Therefore, Mujeres Unidas integrated community mobilization measures [14,15,17,19] into the Rhodes’ Risk Environment framework [34,35] to examine the prevalence and potential of CM in the micro-and macro-level physical, social, economic, and policy risk environments facing women in the sex trade in Tijuana, Mexico. Until now, the potential for women using substances in the sex trade to engage in CM and ABCD approaches in this US-Mexico border region had not been demonstrated. We argue that the exclusion of women using substances in the sex trade from services and organizations should be re-examined through the contexts of risk environments and ABCD approaches. Therefore, we aimed to examine the prevalence of and potential for community mobilization and its association with HIV/STI risk, substance use, and violence victimization among women in the sex trade in Tijuana, Mexico.

## 2. Methods

### 2.1. Study Setting

Tijuana, Mexico (population 1.5 million) [36], is located along the busiest land border crossing in the world and has been a site of escalating HIV prevalence (1%, nearly triple the national average of Mexico) [31]. In the context of this border city, an estimated 9000 women are in the sex trade [32]. While HIV rates have declined globally, they increased six-fold for women trading sex in Mexico [37,38]. The risk for HIV among women in the sex trade was 35× greater than that of other women, aged 15–49, in Mexico, the highest among all Latin American countries [37].

Mexico does not have a universal policy legalizing the sex trade. In Tijuana, Baja California, the Municipal Health Directorate (DMS) is in charge of regulating women in the sex trade, issuing an electronic Health Control Card renewed monthly [39]. Registration allows the women to work without being harassed or persecuted by the police in fixed establishments such as bars and massage parlors in a specific section of the city. However, as many as 50% of the women in the sex trade go unregistered [40] due to a variety of reasons, including the costs of registration and medical fees at government-run STI clinics and undocumented immigration status. Registration costs $1,641.00 MN (121.55 USD, TC 13.5) with monthly fees of $328.00 MN (24.29 USD, TC 13.5) [41] for the application of HIV and STI screening tests. Of the estimated 9000 in 2006, 4850 were registered [42]. However, more recent health control records report a decrease in the number of women with an active card, ranging from 4079 in 2012 to 2841 in 2014 [41].

#### Research Study Sites

To examine these healthcare and safety gaps, Mujeres Unidas (funded by the U.S. National Institute on Drug Abuse, K01DA036439 PI: Urada) employed a mixed-methods design, embedded in Proyecto Mapa de Salud’s (also funded by NIDA, R01DA028692, PI: Brouwer) longitudinal (baseline, 6-, 12-, 18-month follow-up) survey study [43,44]. The latter focused on HIV and substance use risks among venue and street-based women in the sex trade in two Mexico/U.S. border cities (Tijuana and Ciudad Juarez). Mapa de Salud defined social, spatial, and physical factors affecting women in the sex trade and determined their relevance to HIV/STI transmission, drug use, and access to services. Mapa de Salud involved two Tijuana offices, one in the red-light district (set up a decade prior) and a second one in the eastern zone of Tijuana. The offices were known by the participants to be a safe place where they could stay to take their routine study tests or interviews.

### 2.2. Study Sample and Data Collection Procedures

To recruit participants, mapping and time-location sampling were used at bars, brothels, hotels, alleys, and street corners in Tijuana [45]. Sites were geographically stratified with a maximum of 15 recruits per venue and roughly equal numbers of street and fixed venue sites. Recruitment days/times were randomly selected based on hours of operation and available times of the participants [46]. Potential participants underwent a five-minute screening at the point of contact and received USD 5. To be eligible, they were 18 years or older; cis-gender female; exchanged sex for money, drugs, goods, or shelter (during four or more days, with four or more clients within the last 30 days); willing to undergo both behavioral surveys and HIV/STI tests; agreed to receive STI treatment if testing positive; and resided in Tijuana with no plans to move permanently outside the city over 18 months of the study. At least 25% were predicted to use alcohol/drugs based on previous studies with this population [47].

Study Procedures. Informed consent, survey, and laboratory testing occurred in the study office in Tijuana, Mexico [43,44]. Participants met with interviewers (experienced outreach workers) who explained the study to the participants and obtained informed consent. Participants then received $30 for the 45-min interviewer-led survey, followed by rapid HIV testing (results ready in 20 min, plus post-test counseling and a second test and an active referral to the local Municipal Health Clinic if tests were positive) and STI testing (syphilis serology, vaginal swabs to test for Chlamydia and gonorrhea, and pap smears). Interviews were translated into Spanish and reviewed for accuracy and cultural and linguistic appropriateness. The local non-government organization (NGO) study partners in Tijuana (*Centro Ser* and others) provided transportation, condoms, HIV/STI information, and drug use prevention and treatment. All measures were administered using CAPI (NOVA software, MD, USA). Participants were asked to return within one month (receiving $5 cash incentives) to receive the remainder of their STI test results and confirmatory HIV results. For active syphilis (e.g., titers >1:8), gonorrhea, or Chlamydia, all women in the sex trade were eligible under Mexico Ministry of Health guidelines for free on-site treatment [48].

All Mujeres Unidas qualitative interviewers (one cis-gender male, one cis-gender female) understood the slang and context of the women in the sex trade in Tijuana and knew the participants from conducting Mapa de Salud (Findings from interviews with the female participants are published elsewhere) [49].

### 2.3. Measures

Mapa de Salud measures, including the Mujeres Unidas CM measures, were from previously validated instruments or went through a series of stakeholder feedback and piloting. To revise the measures, CM measures were cognitively tested with 10 participants in conjunction with the computerized survey (lasting 60–90 min). Cognitive interviewing is a qualitative approach to pre-testing a survey instrument for an item’s cognitive validity [50] to ensure readability and acceptability. Participants were probed after completing the section with the new mobilization items to determine if they were comprehensible, or had any recall, social desirability (decision processes in answering the question) issues, and how responses coincided with the response options (response processes).

We adapted the existing CM measures (e.g., collective identity, efficacy, and agency) to the Tijuana context, based on the Risk-Environment-Framework-generated community mobilization variables (Figure 1), utilizing themes that emerged from a formative phase. Figure 1 displays the community mobilization micro-and macro-level factors in four types of risk environments that may be associated with the individual risks of substance use, violence victimization, and HIV/STIs. CM concepts of collective identity, efficacy, and agency, as part of “collective power”, were derived from Blankenship et al. (2008) [14] who used a structural interventions framework to analyze the associations between power and condom use among 812 women trading sex and how exposure to a local community mobilization intervention in Andhra Pradesh, India affected these associations. Community empowerment concepts such as (1) decreasing perceived powerlessness and insecurity, (2) increasing access and control over material and social resources, and (3) facilitating social acceptance were adapted from Kerrigan et al. (2008) [17] (originally adapted from the Sonagachi project in India which mobilized women in the sex trade to have power and control over their resources [15,19]).

The outcome measure was community mobilization (peer support via group, meeting, or march where women in the sex trade share concerns, help each other, and/or have goals to take collective action). Other measures of CM were collective identity (a developing sense of community among them); collective efficacy (coming together to address problems); and collective agency (advocating for others, social cohesion, mutual aid) [14]. These parallel with Blankenship’s use of CM and Kerrigan’s increasing social participation as an indicator of CM in the Social Risk Environment.

Several exposures were measured. For the Physical Risk Environment: Decreasing perceived powerlessness and insecurity, macro-level measures included the presence of meetings or workshops among women trading sex, and witnessing violence where they worked. At a micro level, locations where the women gather with others (e.g., venues, public or private street locations) were an exposure, with living and working in the same location as a confounder. For the Economic Risk Environment: Increasing access and control over material and social resources, measures included how often the women could get free condoms and clean syringes and health insurance status. For the Policy Risk Environment: Facilitating social acceptance for CM included having talked or worked with peers in the sex trade to change a situation, believing women in the sex trade can work together to speak up for their rights, and perceiving a right to a life free from violence.

Individual risk environment measures were: Violence and power included the Sexual Relationship Power scale developed by Pulerwitz et al. [51], childhood sexual abuse, intimate partner violence, and violence from clients (ever, recent) [52,53,54]. *Sex risk behaviors* included age at initiation into the sex trade and current patterns of trading sex (e.g., number and frequency of unprotected vaginal and anal sex with clients)*. Substance use* measures included types of alcohol/drugs used in the past month (e.g., cocaine, heroin, methamphetamines, excluding marijuana); route of administration (e.g., injecting, smoking, inhaling); frequency and volume (binge/dependence), and HIV-related substance use risk behaviors in the context of others (e.g., their use of alcohol and specific drugs preceding and during sex with regular, casual and client partners over the past month) [55,56,57,58]. Other measures included injection drug use in the past six months (syringe re-use, receptive/distributive syringe sharing). We created a substance use variable that combined heroin, cocaine, or methamphetamines over the previous 6 months. STI and HIV testing were conducted at the study site clinic (described above).

### 2.4. ABCD Methods

#### Interviews with Service Providers

Using an asset-based community development approach, the researchers also conducted qualitative in-depth interviews with 16 nongovernment and government health providers across 11 sites (e.g., health department, STI clinic, substance use treatment facilities) in Tijuana, Mexico. The sample size was deemed sufficient to reach saturation of themes generated by these interviews.

The interviews aimed to understand these organizations’ roles with hopes of cultivating existing community assets that could be further developed to assist the women. The second author, a native of Tijuana who coordinated the Mapa de Salud and Mujeres Unidas projects in Tijuana, recruited the participants and introduced the University of California San Diego Principal Investigator (PI; first author) to them. Together they interviewed the participants in Spanish and English, using an interview guide with questions/prompts about community mobilization that, like the surveys, were pilot tested first. The PI (also a licensed clinical social worker) and the bilingual research staff were skilled in in-depth interviewing techniques and trained in HIV and ethical study protocols. Research staff met with the service providers in a private space at their offices for in-depth, face-to-face, semi-structured interviews, lasting 60–90 min.

In-depth interview questions included questions about how their organizations served women in the sex trade, especially those using drugs, and whether they were aware that the women in the sex trade in Tijuana had collective agency (social cohesion and mutual aid). Questions included: Are you aware of organizations of women in the sex trade in Tijuana? Have they mobilized or met together to address the problems they face? How about those who use substances? What have they accomplished? What could community mobilization offer?

Data were compared between and across the targeted participants—key informants from the community, government, and consumers. Memos were written which reflected the questions, concerns, and analytical insights emerging from the analyses and served a vital data-reduction and analytical function. Coded qualitative data were used to inform the adaptation of CM measures for the 18-month follow-up survey. This formative research also triangulated and contextualized quantitative findings of this study to give further context to this study.

### 2.5. Data Analysis

Our quantitative data analysis was guided first by our framework (Figure 1) and then, within that framework, we examined measures that had the strongest association with the outcome. We used the Rhodes’ Risk environment framework for the analysis of the Mapa/Mujeres Unidas data. Data were analyzed using bivariate and multiple logistic regression methods guided by Westreich and Greenland (2013) [59] to display the exposures significantly associated with CM. Univariate models were constructed with each exposure, with relevant demographic, health, risk history, and risk environment variables to identify potential confounders. Multivariate models were developed using a manual stepwise procedure whereby variables hypothesized a priori that are associated with HIV-related outcomes along with potential confounders having a significance level < 10% in univariate regressions were considered. To determine the most parsimonious model, adjusted models were compared using the likelihood ratio statistic.

Additionally, we used the asset-based community development model by holding empowering community advisory board (CAB) meetings with the women and conducting qualitative interviews with service providers to look for community assets. Qualitative data from the interviews with service providers, particularly the perception and treatment of those using substances, contextualized the survey results; they were analyzed using a thematic analytic approach and inter-coder agreement [60]. We used participant observation methods for the group meetings and content analysis of the sessions.

## 3. Results

Service providers and women in the sex trade who don’t use substances often exclude women who do. Therefore, the potential of those using substances was examined in terms of engaging and empowering them in CM and ABCD approaches at the community and service provider level. The following results show (1) they are worse off than women who don’t use substances in the sex trade (2) they are less likely to mobilize, and (3) they can mobilize; some intervention facilitators even sustained the CM intervention beyond the study.

### 3.1. Survey Results

A sub-analysis at baseline (*n* = 302) of women using substances vs. not using substances in the sex trade revealed 14 significantly poorer social and structural conditions experienced by those using substances. They were more likely (*p* = 0.001) to have no health insurance; fear police abuse/harassment; never have registered in the sex trade and to feel it was too expensive to do so; be arrested; have felt unsafe at home; experience abnormal genital itching; have had police ask them for money or forcibly took their money. At *p* < 0.05, the women using substances in the sex trade experienced significantly more sexual coercion by police in the last 6 months and were more likely to be robbed by a client or to have a client fail to pay them.

At an 18-month follow-up survey, 13% of the women trading sex (*n* = 26 of 195) said they participated in community mobilization activities (Table 1). Of these, 39% of the women who participated in CM activities used substances. Forty-three percent of the entire sample used heroin, cocaine, or methamphetamines over the previous 6 months. Thirty-nine percent said they never used condoms in the past 90 days with regular clients, 45% said they didn’t have health insurance, and 25% said condoms were never free. Forty-six percent witnessed violence where they worked, 23% could not perceive a right to a life free from violence, and 13% said they were physically abused in the past 6 months. However, 61% talked or worked with peers in the sex trade to change a situation and 51% believed women in the sex trade can work together to speak up for their rights.

In adjusted analyses (*n* = 135 who traded sex in the past 6 months at the 18 months follow up), participation in CM activities (*n* = 26) was more likely among women trading sex who did not report substance use vs. those who did (AOR: 4.36, CI: 1.11–17.16), with perceiving a right to a life free from violence (AOR: 9.28, CI: 2.03–59.26), with talking or working with peers in the sex trade to change a situation (AOR: 7.87, CI: 2.03–30.57), witnessing violence where they worked (AOR: 4.45, CI: 1.24–15.96), and with condoms being free (AOR: 1.54, CI: 1.01–2.35), adjusting for whether a woman lived and traded sex in the same location (Table 2). Involvement in CM-type activities was not significantly associated with reduced risk for women trading sex in terms of HIV/STI behaviors (unprotected sex, sharing syringes), HIV/STI infections, or with less violence victimization.

### 3.2. Qualitative Results

Qualitative interviews with the service providers as well as participant observation methods during the CAB group meetings triangulated the survey measures to gain a deeper understanding of the physical, socio-political, and economic risk environments associated with CM barriers and facilitators.

#### 3.2.1. In-Depth Interviews

At baseline, women had poorer outcomes if they used substances. Therefore, we used an ABCD approach to interview service providers in Tijuana to determine the barriers and facilitators of the providers and local institutions to engage in ABCD with the women.

In-depth interviews with 16 Tijuana service providers across government, non-government, and substance use facilities revealed how women who used substances especially in the sex trade were often perceived in government and society. Drug stigma and misinformation were so strong that some government providers could not perceive the agency of those using drugs. They often believed to make any improvement, the women must “completely stop consuming drugs and even be locked down against their will”. The following quotes illustrate the barriers to mobilization and empowerment that a lack of non-stigmatizing rehabilitation and health/mental health services for women with problematic substance use presents. The social stigma of drug consumption along with a lack of mental health services for them may delay or make mobilization difficult.

Thematic data analysis of these qualitative interviews with service providers revealed that women with substance use issues were often “referred out” to rehabilitation or only offered condoms instead of actively engaging them in ABCD activities.

##### Barriers to Services for Those Using Drugs

They hardly go out, because they know they are not accepted by society… they have a fear of how they would be treated. (Government provider)

Few accept the use of drugs… when you fill up a questionnaire it’s like “Any drug even marihuana you had smoked.” But they won’t share that information; they don’t tend to ask for help. (Government provider)

##### Women with Substance Use Issues “Referred Out” to Rehabilitation Only

They came to the law department, because they always come to the law department, never to psychology… then if we see something wrong, we call the psychologist and in that moment psychologist and lawyer… [we] realize there’s an addiction and we make the recommendation to go to a rehabilitation center. (Government provider)

The problem is that it is voluntary, I mean you go in and if you want to go out, you leave. I mean they say it’s really good, but I believe the process is not as good as being locked in, in comparison to leaving whenever you want. (Government provider, referring to substance use rehabilitation)

It’s just one opportunity… you can’t be giving the opportunity [to enroll in a rehab center] to people that don’t want to be cured, well that don’t want to [make the] effort. The thing is that even if I had a place I wouldn’t give it to her, I would wait for someone who really would like to do something. (Government provider)

##### Women Trading Sex Are “Only Offered Condoms” Instead of Other Assistance

Women in the sex trade are sometimes “only offered condoms” when they go to a government office. (Non-profit provider)

There is, what can we say? Discrimination, lack of attention, or trained people, we see these vulnerable population, truly suffering because of everything. Who treats this vulnerable population? No one, not even one institution. Then what they do is to hide and show up when they are pretty sick, when there’s nothing they can do…with a complex diagnosis and only that way they’ll receive care. (Government provider)

Speaking at a state level, there’s still a belief that the drug user could be equated to a mental alienated. There is the problem… then it gives the appearance that they don’t have control of what to do with their lives. There’s when the human rights of the person are violated. (Non-profit provider)

Though medical and government assistance seemed limited for women with the most need, some of the service providers offered to attend a Mujeres Unidas CAB meeting to find out what the women needed. In this way, we drew upon the existing assets and potential capital of these service providers.

#### 3.2.2. Results from the Mujeres Unidas CAB Meetings Held with Women Injecting and Non-Injecting in the Sex Trade

Following the interviews and initial surveys, we set up space and designated times to meet as a group with the women who used substances in the sex trade. We did not pay them cash, but instead gave them food and beverages, hand lotions, backpacks, arts and crafts materials to use in the group while they talked about topics such as HIV, substance use, and violence. We conducted monthly 3-h CAB meetings with a total of 45 participants (women in the sex trade who used substances, injecting, and non-injecting) who attended at least one meeting over a year. The purpose of these groups (with 10–15 attending at a time) was to validate the research results and to hear their concerns, as well as to examine their strengths/assets and potential to gather.

The Mujeres Unidas PI elicited the support of a primary group facilitator (fourth author on this manuscript), a Tijuana medical doctor (MD) the participants already trusted because of her role as the Project Director of another multi-year UC San Diego NIDA-funded HIV research project El Cuete (PI: Strathdee, funded by NIDA, focused on those who inject drugs). This MD group facilitator owned the space used for the CAB meetings (Casa del Centro, a retreat center located close to the areas where the women worked). We also relied on a bilingual peer research staff outreach worker who recruited the participants directly from the street and also helped facilitate the groups. Two more field staff/coordinators from Tijuana (male and female) participated in developing and facilitating CAB content. The PI also delivered the research results in Spanish to the participants and was present at all meetings in a participatory observational way (bringing food and serving it to break down the barriers of being an academic from the US).

The CAB meetings included exercises in Spanish to help the women voice their concerns with each other and the group facilitators. The peer outreach worker shared her experience living with HIV and substance use recovery, local clinic medical providers facilitated discussions to understand the participants’ needs better, and other research staff/interns led group exercises to help the women explore their goals and strengths. All groups were conducted in Spanish. Offering these meetings helped us have a basis in the 18-month survey to compare those who participated in mobilization-type activities vs. those who had not.

##### Themes from the CAB Meetings

The women said they liked the space because it “took them away from their routine and tough life in the sex trade.” The location, Casa del Centro, brought back memories of a home from their hometowns- They said it “felt like a house.” The facilitator, a medical doctor who owned the space, recalls one woman, “Ale,” who passed away one or two months after the group ended.
She was halfway out [from substance use] but when she was awake, she was saying, looking at her painted wooden box [done during the group meeting]: ‘Is not bad, actually, I like it. I did not know that I could do something pretty with my hands! (“No esta mal, hasta me gusta, no sabia que podia hacer algo bonito con mis manos!”)

Domestic violence from partners, pimps, and clients was a major theme. When a woman shared and asked for help, some of the women were supportive and gave her advice. A clinic and peer outreach staff from the research team showed respect and trust to the people who came to the group. This made the participants feel grateful that we took the time to provide the talks and space because they usually did not have time or didn’t feel comfortable or their checkups with their doctors were too quick for them to talk to their doctors about the issues they faced or questions they had.

When we talked about getting a letter to the Secretary of Health about their needs, the women in the group were enthusiastic about doing something. The medical doctor who facilitated the group presented the wishes of the women at a binational conference attended by the National Human Rights office. The top three desires the women had for changes in government policy were: less expensive health care, more protections against abuse for safety, and changes in police practices. These desired changes came up in the study’s 12-month follow-up survey results (Table 3) and were validated in the CAB meeting discussions with the 45 attendees. Other desired changes were less discrimination against those using substances and in healthcare, better STI clinic hours, drug treatment, and the rescue of those trapped in the sex trade.

#### 3.2.3. Reflexivity of the Positionality of the Researchers

Coming from the U.S., the PIs did not have the same cultural, linguistic, nor social understanding of the Tijuana context which is why care was taken to incorporate into all stages of the research those with lived experience and those who were native to Tijuana and already connected to the population. Those bilingual in Spanish and English and living in Tijuana helped to co-facilitate the interviews and the CAB meetings and interpret and draw conclusions from the data. Finally, an author with lived experience as a human trafficking survivor in the sex trade in Tijuana provided consultation on this manuscript. The women in the study also welcomed the male facilitator of the research interviews and CAB meetings because of his positionality as a gay male and respected former NGO member.

## 4. Discussion

This study broadens previous community mobilization models and the Rhodes risk environment framework to encompass barriers and enabling factors for CM in the context of four risk environments faced by women in the sex trade, particularly those who use substances, in Tijuana, Mexico. The results indicated that the women in the sex trade participated in community mobilization less if they used substances, but more so if they witnessed violence at work or believed in a right to a life free from violence. However, the triangulated results of qualitative interviews with government and non-government service providers and the women using substances who engaged in Community Advisory Board meetings demonstrated a potential for asset-based community development to occur for those who are the most marginalized. The women using substances found community and voice with each other, despite common beliefs that injecting and non-injecting women could not come together. As part of asset-based community development, service providers began to engage with the women during these CAB meetings.

In terms of the integration of CM into Rhode’s risk environment framework, the results indicated that the physical risk environment—witnessing violence where they worked—may motivate women in the sex trade to address their human rights with others. Likewise, in the socio-political risk environment, talking and working with peers to change a situation, and believing in the right to a life without violence, were associated with greater participation in community mobilization. These findings coincide with other studies of women in the sex trade who have experienced violence or police abuse, and at times mobilized against it [61,62,63,64,65,66,67]. On an economic and individual level, those who mobilized tended to have more access to free condoms, perhaps indicating decreased isolation and greater access to knowledge and resources about HIV prevention.

However, understanding the needs of women in the sex trade beyond HIV prevention and condom distribution can be an important step for delivering more effective structural interventions, especially where women have few resources to stop their substance use or sex trade involvement on their own. Women expressed the need to help those trapped in the sex trade. For those using substances, community mobilization was successful among those in India and the United States, e.g., using peer educators to disseminate information on needle syringe exchange [68,69]. Such strategies can assist women with recovery when they need it [70].

Findings from the qualitative interviews with service providers illustrate both barriers and potential for asset-based community development to occur in this community. Government and non-government agencies at the municipal and state levels could address the human rights of this vulnerable population through a collective impact model [71]. Such a model would bring together sectors such as substance use providers, women’s rights and domestic violence agencies, child welfare, homelessness agencies, HIV and medical clinics, police, and human trafficking organizations to address the violence, substance use, and health risks these women face. Such efforts could address the isolation, stigma/discrimination, and abuse the women endure despite the registration of women in the sex trade and mandated government clinic STI check-ups. Women could have a seat at the table such as the human trafficking survivors who now assist others exiting the sex trade in San Diego County, U.S.A.

## 5. Limitations

Aside from the biological data collected on HIV/STIs during the survey study, this study depended on recall and self-report of sensitive behaviors. However, in substance use research, test-retest reliability for sexual behaviors has been documented as good (0.62–0.90) [58], suggesting that accurate recall of sexual behaviors for the past month does not impact the results. Attrition occurred in subsequent time points of this study among those who participated in the Mujeres Unidas-led CAB meetings. Additionally, women who belonged to an existing organization of women in the sex trade in Tijuana did not participate in this study. Therefore, the prevalence of community mobilization among all women in the sex trade does not reflect the activity of members of that organization. Transgender women, although overrepresented in the sex trade and among those living with HIV, also did not participate in this study. Tijuana research studies were typically separated by cisgender and transgender identities. Future CM group meetings could include Transgender individuals. Likewise, the responses from a cross-section of service providers (government and non-government) who participated in this study do not necessarily reflect the perceptions of all service providers in Tijuana, Mexico, nor of the current political appointees that change over time. However, the attitudes towards women using substances in the sex trade especially may prevail as long as the rehabilitation models, practices, and policies towards them do not evolve to encompass a more strength-based or ABCD approach.

## 6. Final Considerations/Recommendations

This study examined the community empowerment of women trading sex in Tijuana, Mexico, corresponding to the World Health Organization guidelines “to reduce HIV infections among women in the sex trade by improving their access to health services based on a human rights approach [69]” as well as the UN sustainable development goals outlined at the beginning of this paper. Building on CM measures by examining the prevalence of CM within Rhode’s Risk Environment framework, Mujeres Unidas discovered that women using substances in the sex trade were less likely to mobilize, yet they faced the greatest harms and were excluded from some organizational assistance. Taking an asset-based community development approach, this study is the first to examine the potential of CM among women using substances in the sex trade in Tijuana, Mexico. In CAB meetings, we discovered that women’s empowerment and health outcomes might be further enhanced by ABCD, beginning with the government and community supporting the empowerment of the women using substances in the sex trade to engage in supportive services with them, instead of just referring them out to government rehabilitation programs or other limited practices [72]. This study demonstrated that women using substances (injecting and non-injecting) in the sex trade could participate in community and government organizational efforts to find community and voice amongst themselves, despite often being excluded by others for their substance use

Applying CM to a Risk Environment framework and ABCD lens may open a door to include traditionally isolated groups of women who use substances in settings where they may be excluded from other associations and services. It humanizes their voices and provides spaces such as Mujeres Unidas to invite previously silenced voices to feel empowered, identify common life challenges, and promote group sharing and motivation to gather as a community or a movement. ABCD can help change the situations of women who use substances in the sex trade by assessing their risk environments and connecting them to untapped potential social capital for treatment and recovery support. CM opens an opportunity to advocate for the inclusion of people who use substances in the sex trade into health and social services. Future research can further draw upon existing community and government assets to help women using substances in the sex trade overcome barriers to getting assistance for the substance use, violence, and harms to their health they face.

## Figures and Tables

**Figure 1 ijerph-18-03884-f001:**
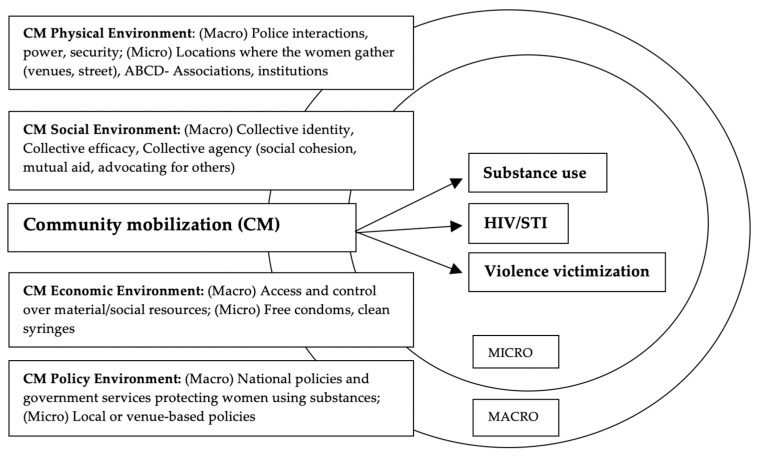
Mujeres Unidas theoretically expands Community Mobilization (CM) models to include CM & Asset-based Community Development (ABCD) in the context of four risk environments that may be associated with individual substance use, HIV/STIs, and/or violence victimization.

**Table 1 ijerph-18-03884-t001:** Characteristics of women in the sex trade in Tijuana, Mexico, and their bivariate associations with community mobilization (*n* = 195), at 18 months follow-up.

	Total *n* (%)	No Community Mobilization (*n* = 169)	Community Mobilization (*n* = 26)	*p*-Value
Physical risk environment				
Living and working in the sex trade in the same location				
No	187 (96)	166 (98)	21 (81)	0.001
Yes	8 (4)	3 (2)	5 (19)	
Witnessing violence where they worked *				0.008
No	73 (54)	68 (59)	5 (25)	
Yes	62 (46)	47 (41)	15 (75)	
Economic risk environment				
How often can you get condoms for free?				0.161
Never	49 (25)	45 (27)	4 (15)	
Sometimes	71 (36)	62 (37)	9 (35)	
About half of the time	10 (5)	9 (5)	1 (4)	
Often	23 (12)	18 (11)	5 (19)	
Always	42 (22)	35 (21)	7 (27)	
Had health insurance				0.171
No	88 (45)	73 (43)	15 (58)	
Yes	107 (55)	96 (57)	11 (42)	
Socio-political risk environment				
Talked or worked with peers in the sex trade to change a situation **				0.006
No	119 (61)	108 (64)	11 (42)	
Yes	75 (39)	60 (36)	15 (58)	
Believes women in the sex trade can work together to speak up for their rights				0.242
Strongly agree	99 (51)	83 (49)	16 (62)	
Agree	30 (15)	27 (16)	3 (12)	
Neutral	20 (10)	17 (169)	3 (12)	
Disagree	8 (4)	7 (4)	1 (4)	
Strongly disagree	38 (20)	35 (21)	3 (12)	
Perceives a right to a life free from violence				0.323
No	45 (23)	41 (24)	4 (15)	
Yes	150 (77)	128 (76)	22 (85)	
Individual risk environment				
Substance use (heroin, cocaine, or meth, past 6 months)				0.650
No	112 (57)	96 (57)	16 (62)	
Yes	83 (43)	73 (43)	10 (38)	
HIV/STIs				0.256
No	165 (85)	141 (83)	24 (92)	
Yes	30 (15)	28 (17)	2 (8)	
Condom use with regular clients (past 90 days)				0.351
Never	76 (39)	69 (41)	7 (27)	
Sometimes	9 (5)	8 (5)	1 (4)	
About half of the time	9 (5)	7 (4)	2 (8)	
Often	13 (7)	8 (5)	5 (19)	
Always	88 (45)	77 (46)	11 (42)	
Used syringes others used				0.361
No	178 (91)	153 (91)	25 (96)	
Yes	17 (9)	16 (9)	1 (4)	
Physically abused, past 6 months				0.408
No	170 (87)	146 (86)	24 (92)	
Yes	25 (13)	23 (14)	2 (8)	

* *n* = 135 for those who traded sex (past 6 months) ** Situations included: Reproductive health problems (e.g., contraceptives, abortion); Abuse or violence from clients; Clients stealing money; Clients refusing to use a condom; Police arrests; Difficulties with police such as violence or extortion; Police taking condoms away; Problems with club or bar management; Rights as workers; Need for better access to health care; HIV/STI prevention needs (e.g., access to information, condoms); Child care or other issues caring for children; Getting paid for sex; Financial problems or economic debts; Getting evicted or housing issues; Problems related to substance use (e.g., police taking syringes away); Problems with getting sex trade registration and paying fees at the Control Sanitario; Relationship problems with boyfriend or spouse (e.g., domestic violence); Health insurance/social security/obtaining other documents/government assistance; Negotiating better pay; Finding other employment and/or stopping sex trade involvement.

**Table 2 ijerph-18-03884-t002:** Exposures significantly associated with community mobilization activity among women in the sex trade in Tijuana, Mexico (*n* = 135) at 18 months follow-up.

Variable	Crude OR	Adjusted Odds Ratio	Confidence Interval
Physical risk environment			
Witnessing violence where they worked (Y/N)	4.34 (1.48−12.76)	4.45	1.24–15.96 **
Economic risk environment			
Getting free condoms (frequency)	1.21 (0.93–1.59)	1.54	1.01–2.35 **
Socio-political risk environment			
Talking or working with peers in the sex trade to change a situation (Y/N)	1.11 (1.20–1.21)	7.87	2.03–30.57 *
Perceiving a right to a life free from violence (Y/N)	1.50 (0.49–4.65)	9.28	1.45–59.26 **
Individual risk environment			
No substance use	1.22 (CI: 0.52–2.84)	4.36	1.11–17.16 **

* *p* < 0.01, ** *p* < 0.05, adjusting for living and trading sex in the same location.

**Table 3 ijerph-18-03884-t003:** Desired changes in government policy, women trading sex in Tijuana * (*n* = 229).

Less expensive health care	45%
More protections against abuse for safety	34
Changes in police practices	25
Reproductive health services (e.g., family planning/contraceptives, post-abortion care)	21
Less discrimination in healthcare against women in the sex trade	18
Less discrimination against those using substances	12
Better hours for clinic appointments	12
Better drug treatment	7
Rescue of those who are trapped in the sex trade	5

* from the 12-month follow-up survey.

## Data Availability

The data presented in this study are available on request from the corresponding author. The data are not publicly available due to privacy restrictions.

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
