# Peer review of "Mujeres Unidas: Addressing Substance Use, Violence, and HIV Risk through Asset-Based Community Development for Women in the Sex Trade"

_ijerph, 2021, doi:10.3390/ijerph18083884_

Round 1

Reviewer 1 Report

Thank you for the opportunity to review this paper, which considers community mobilization of sex workers—a very important topic. While the paper draws on what seems like a very extensive and interesting study, it is not yet ready for publication in its current form. Below I offer some suggestions for how the authors may clarify their key concepts and theoretical framing to ensure their findings speak to a wider audience.  

Key Concepts: ABCD & CM

I encourage the authors to situate more explicitly, for the reader, the broader literature in which ABCD and Community mobilization (CM) are located. What discipline do these concepts come from? For example, are they related in some way to political science studies of civic engagement, or collective action?

I also ask this because the authors write at line 66, “few studies until now have examined the potential for substance-using women trading sex to also participate in ABCD and CM approaches.” However, there have been studies of how sex workers are mobilized through their communities, through HIV/AIDS prevention organizations (see eg Majic, Samantha. 2014. "Political Participation Despite the Odds: Examining Sex Workers’ Political Engagement."  New Political Science36 (1):76-95.; Stoller, Nancy. 1998. Lessons from the Damned: Queers, Whores, and Junkies Respond to AIDS. New York: Routledge.) So in what way is the CM that the authors discuss different from what is happening in these (and many other) examples)?

To clarify their concept of CM, the authors also need to engage more with work about the sex worker rights movement. They write at line 75, “For women in the sex trade, community mobilization might mean mobilizing for health, safety, and human rights via a group, meeting, or marches where women share concerns, help each other. …” Scholars and activists have written extensively how sex workers of all genders, and women specifically, have mobilized and empowered their community re. HIV/AIDS prevention, occupational health and safety, etc *without* an explicit ABCD/CM orientation (see the following (for just some examples Majic, Samantha. 2014. Sex work politics : from protest to service provision. First edition. ed, American governance: politics, policy, and public law. Philadelphia: University of Pennsylvania Press; Chateauvert, Melinda. 2013. Sex workers unite : a history of the movement from Stonewall to Slutwalk. Boston: Beacon Press; Shah, Svati Pragna. 2014. Street corner secrets : sex, work, and migration in the city of Mumbai, Next wave : new directions in women's studies. Durham: Duke University Press.)Given all of this existing literature, the authors need to clarify how their research differs here or, put differently, what does a CM orientation add that other studies of sex worker mobilization have missed?

Theoretical frame

The theoretical frame needs a lot of development, in a separate section, especially for a journal like this that has a broad readership. The authors write at Line 100 “Using Rhodes’ Risk Environment theoretical framework …” What is this framework, for non-specialists, and why is it the best for this paper? Later, the authors write “Community mobilization domains and measures, including the concepts of collective identity, efficacy, and agency, were derived from Blankenship et al (2008) [5] and Kerrigan 232 et al (2008) [8], and community empowerment concepts from Swendeman et al (2009) [16].” Again, what are these theories? What do they say about CM? And what is the theoretical gap in all of them that your research will fill?

Case selection & Methods

My main question about the study procedures relates to my initial comments re. defining ABCD/CM: how did the researchers “do” ABCD? There are numerous statements to the effect of “Using an asset- based community development approach …” and “Thirty-nine percent of the women trading sex (n = 26 of 169) participated in community mobilization activities. ...” But there is little explanation of what this approach (ABCD/CM) looked like in practice, in the study: the authors discuss CM as something to be measured (p.3) but it also seems like it could be an activity they did with the participants. What is it?

It’s not until Line 372 that the reader gets some idea of what ABCD/CM may involve, when the authors write “the substance-using women who participated in the Mujeres Unidas community advisory board meetings …” Was this—community advisory board meeting participation—how the study did and/or conceived of CM? Detailing, early in the paper, how the authors did this (ABCD/CM) is key, since its central to the study. How did the researchers get participants to develop assets, for example? What were the CM activities used? Without this information, is it impossible for the reader to understand how the community was empowered.

Regarding context, the authors must also provide more information to justify their case: why Tijuana? Also, what is the legal status of sex work in Tijuana/Mexico more broadly? This information re. legal context is important, as legal status is a key structural factor that can shape health outcomes (i.e. criminalization is bad for sex workers’ health).

Say more about Mujeres Unidas” and “Proyecto Mapa de Salud” --  there is some detail at line 166, but for readers not familiar with the studies, more detail would help-- who formed these? Who funds them? Who were the “NGO partners” (line 194)? Were these research projects a collaboration with the sex workers in Tijuana in some way? Also, the authors could do more to position themselves in relationship to the community studied. How were they reflexive about their positionality (eg as academics from the US, in cases) in relation to the sex workers they interviewed in Mexico? Did everyone speak the same language? What kinds of questions did the survey cover for this study?

Findings

The discussion of how were perceived (line 312+) was interesting but a bit “out of nowhere” – how does the material here relate to and support the broader argument re. ABCD/CM? It seems the interview data from the sex workers would be more relevant to the paper/argument to illustrate how they felt/were empowered by the ABCD/CM (in whatever form that took).

Line 68 Although substance use is high among women in the sex trade … Please cite this or clarify it, as sex workers, like the rest of the population, vary in their substance use.

Conclusion

I suggest that the author’s re-work the conclusion as follows: discuss 2-3 contributions to whatever broader area of theory they engage (eg Blankenship framework?)-- Once the paper clarifies this theoretical framing, this will be easier to write; 1-2 ways their study expands issue-specific knowledge (eg community mobilization, sex worker health and safety), and 1-2 directions for future research.

Thank you for the opportunity to review this very interesting paper!

Reviewer 2 Report

Interesting manuscript but that can improve with some revisions.

1. The abstract is huge, does not follow the rules of the journal and must be revised. A maximum of 200 words is more than necessary;

2. The introduction is also huge and repetitive. Please bear in mind the role of the discussion when presenting the research problem and the reasons why it is interesting to be researched. Stick to answering:
-What is the prevalence of HIV and STIs in female sex workers? What are its determinants? what is community mobilization (MC) and how can it help solve this problem? You say that little has been published on the subject and that would be the knowledge gap that would justify the originality of your paper. However, for this it is necessary that you present a review of the current language that corroborates your phrase or even that the authors search for systematic data that may subscribe to such information.

3. Acronyms must be explained the first time they appear in the text

4. In view of the objectives, it is redundant to present hypotheses.

5. Contextualize your work in global health; agenda 2030 and SDGs (UN).

6. Regarding the questionnaire:

-Describe who made the instrument

-What is the theoretical framework that supported its construction;

-Describe your variables, type of questions, way of evaluating responses

- Was there a validation or pre-test? If so, how did the instrument's validation process take place (content, clarity and objectivity) and what technique was used for consensus in the questions. This is very important since most of these questions have instruments validated in the literature for this subject. In the absence of a validation process, the research is subject to measurement and interpretation views.

7. The method of entering variables in the final model of logistic regression only by the p-value is debatable in the literature.

8. In table 02, please add the gross odds values.

9. I suggest a reorganization of table 02. It is not clear the reference variables. Gross Odds should be included so that we can understand where we started from and where we went. Furthermore, the high confidence intervals are doubtful and seeing the gross odds can help to understand this.

10. Something that made me very uneasy at work is the lack of integration of qualitative and quantitative data. There is no real integration and that leaves the paper poor. What theory underpinned this integration? In addition, it is necessary to clarify, with regard to qualitative data,:

-Which author / s conducted the interview or focus group?

-What were the researcher's credentials? 

-Was the researcher male or female?

-What experience or training did the researcher have?

-Was a relationship established prior to study commencement?

-What did the participants know about the researcher?

-What characteristics were reported about the interviewer/facilitator? 

-What methodological orientation was stated to underpin the study? 

-Were questions, prompts, guides provided by the authors? Was it pilot tested?

Round 2

Reviewer 1 Report

Thank you for the opportunity to review this revised paper. I think that the authors have done significant work to improve the paper, but I still believe it requires more work, particularly to clarify the paper's main argument, presentation of findings, and theoretical contributions.

First, please shorten the abstract further!

Second, the intro still requires some work. I see that the other reviewer asked for the SDGs, which seems relevant, but how do they connect to ABDC? Altogether, the material to approx. line 70 that discusses ABCD and the SDGs is hard to follow and reads as though the SDG material was just pasted in. Consider finessing somehow to clarify the SDG-ABCD link.

In the intro, the purpose of the study is now clearer (lines 88-97), but what is the main argument? What is the overall outcome of the examination? What should readers take away/learn from this? I.e. what are the results (line 351+) illustrating? The results section should then be revised to more clearly connect the results to the argument-- what parts of, specifically, are they illustrating?

Lines 123-124: insert a transitional sentence. How do the stats provided in the paragraph to line 23 connect to the info about Mujeres Unidas in the next para?

The methods section is much more detailed now but extremely long. Some clarification about the following would be helpful:

Why were only “biologically female” women included? (line 152). Do you mean cisgender women? And why the exclusion of trans women? Please provide some explanation here, given the fact that trans women are at higher risk for exploitation in the sex industry/

Overall, I appreciate the extra details (requested) in the methods discussion, but it now encompasses ½ of the paper. Is there any way to make this more concise and, in places, less repetitive? Possibly some of this material could be incorporated in the results section?

Conclusion:

The authors claim in their response to have explained their paper’s theoretical contributions in the conclusion, but I don’t see this (presumably this would be in lines 547+). This section mainly, still, re-summarizes the paper and just adds directions for future research. Consider revising more in line with my earlier suggestions.

Thank you for the opportunity to review this paper again.

Reviewer 2 Report

Congratulations on the extensive corrections in the manuscript! I suggest:

-Abstract must have a aims and be around 250 words;

-Please keep only general objective in the text;
